# Comorbid Hypothyroidism and Low-Alanine Aminotransferase-Associated Sarcopenia Associated with Shortened Survival: A Retrospective Study of 16,827 Patients over a 21-Year Period

**DOI:** 10.3390/jcm13195838

**Published:** 2024-09-30

**Authors:** Omer Segal, Rabia Khoury, Adva Vaisman, Gad Segal

**Affiliations:** 1Internal Medicine “I”, Chaim Sheba Medical Center, Faculty of Medical & Health Science, Tel-Aviv University, Tel Aviv-Yafo 6997801, Israel; omer.segal@sheba.health.gov.il (O.S.); adva.vaisman@sheba.health.gov.il (A.V.); 2Education Authority, Chaim Sheba Medical Center, Faculty of Medical & Health Science, Tel-Aviv University, Tel Aviv-Yafo 6997801, Israel

**Keywords:** hypothyroidism, sarcopenia, frailty, survival, alanine aminotransferase, prognosis

## Abstract

**Simple Summary:**

True personalization of internal/general medicine depends on the assessment and acknowledgment of sarcopenia and frailty characteristics of patients. The co-existence of sarcopenia, frailty, and hypothyroidism must be acknowledged by physicians, since these are common comorbidities bearing negative long-term clinical outcomes. Low ALT values could serve as a useful biomarker for screening of patients already diagnosed with hypothyroidism.

**Abstract:**

**Background:** Hypothyroidism is very common worldwide. It is known to be associated with frailty which, in turn, is associated with increased morbidity and mortality in the elderly. Low ALT blood activity is an established marker for sarcopenia and frailty. The incidence and outcomes of the association between low ALT values and hypothyroidism, as manifested in elevated blood TSH levels, is unknown. The objective of this study was to assess if low ALT values could improve the prediction of clinical outcome in hypothyroid patients. **Methods:** This was a retrospective analysis of hospitalized patients in a large, tertiary hospital. **Results:** Over a period of 21 years, an overall population of 16,827 patients were identified as eligible to participate in this study. Within the study population, 726 (4.3%) were classified as suffering from hypothyroidism (TSH values > 6 MIU/L) and 2184 (13%) were classified as patients with sarcopenia (ALT < 12 IU/L). Within our patient population, hypothyroidism was associated with sarcopenia in a statistically significant manner (*p* = 0.011). Patients classified as suffering from both hypothyroidism and sarcopenia had significantly shorter survival: A multivariate analysis showed that the frail and hypothyroid group of patients had a statistically significant risk of mortality in the next 5 years (HR = 3.6; CI 2.75–4.71; *p* < 0.001). **Conclusions:** Sarcopenia and frailty are common comorbidities, bearing negative long-term clinical outcomes. Low ALT values could serve as a useful biomarker for screening of patients already diagnosed with hypothyroidism.

## 1. Background

### 1.1. Sarcopenia and Frailty Negatively Impact the Longevity of the Mid-Life and Elderly Population

Frailty is an aging-related syndrome that renders patients more vulnerable to health-related risks, and a significant volume of clinical research is aimed at this realm. Ever since Fried et al. presented their “evidence for a phenotype” in the year 2001 [1], numerous studies engaged definitions and subtypes of frailty, with disability being the inevitable result of this rather new syndrome. The major bulk of literature, however, presented frailty as a common phenomenon amongst community-dwelling elderly people. The rate of frail, but not necessarily old, patients within hospitals is less well known and researched. Nevertheless, a retrospective cohort study that was conducted on elderly patients hospitalized for Obstructive Sleep Apnea (OSA) showed that the higher the frailty score was, as defined by the Hospital Frailty Risk Score (HFRS), the greater the patients’ risk of mortality. This study and others have proven the strong association between frailty and subsequent mortality of hospitalized patients [2]. A cross-sectional, longitudinal study was conducted to evaluate the potential effect of sarcopenia and frailty on morbidity and mortality among peritoneal dialysis (PD) patients. In this population, frailty and sarcopenia were determined using the Clinical Frailty Score (CFS), and the Asian Working Group for Sarcopenia criteria, respectively. In this study, once again, a positive correlation between sarcopenia, frailty, and mortality was found [3]. Tanaka and co. conducted a prospective study in which elderly individuals were assessed based on their oral frailty (according to the following factors: number of natural teeth, chewing ability, articulation, tongue pressure, and subjective eating and swallowing difficulties) over a 4-year period (spanning from 2012 to 2016). They found a strong correlation between oral frailty at baseline and future frailty, sarcopenia, and mortality [4]. Pre-frailty and frailty appear to affect middle-aged as well as elderly patients, suggesting that more spotlights should be directed to middle-aged frail patients [5].

### 1.2. Hypothyroidism Is Potentially Associated with Sarcopenia and Frailty

Hypothyroidism is considered as a background cause for exhaustion (alongside other pathophysiology such as depression, anemia, hypotension, and vitamin B12 deficiency), and therefore it is recommended by clinical guidelines to be ruled out, as part of the expected management of patients diagnosed as suffering from sarcopenia and frailty [6]. Virgini and co. conducted a cross-sectional, prospective study aimed at assessing the potential association between sub-clinical hypo- and hyperthyroidism with different levels of frailty and sarcopenia. The frailty was determined by using a modified Cardiovascular Health Study Index and their results did not establish such association [7]. Abdel-Rahman, Mansour, and Holley questioned the potential contribution of hypothyroidism to the frailty status of elderly people suffering from chronic kidney disease. Moreover, they hypothesize that supplementation of thyroid hormones could, potentially, treat frail elderly people [8]. 

### 1.3. Low ALT Values Serve as a Biomarker for Sarcopenia and Frailty

Alanine aminotransferase, catalyzing conversion of pyruvate to the amino acid alanine, and vice versa, taking part in carbohydrate metabolism [9], plays a major role in several metabolic pathways in various tissues like the liver and muscles. As it is an intracellular enzyme, its blood levels are most commonly used to trace and monitor liver tissue damage in a wide range of conditions, such as viral, toxic, and ischemic hepatitis. In recent years, evidence has accumulated, pointing out the clinical circumstances in which the ALT blood levels are low—a laboratory finding that was generally overlooked. It was shown that, in the absence of liver tissue damage, blood ALT levels serve as a reliable marker for the whole body’s striated muscle mass. Therefore, low–normal ALT levels, when associated to gender and age, serve as a biomarker for decreased muscle mass and functionality, namely—sarcopenia and frailty. 

Epidemiologic studies show that while normal ALT levels proved to be protective against overall and cardiovascular mortality [10], patients with low serum ALT levels with a myriad of clinical diagnoses in their background had a high prevalence of sarcopenia and frailty [10]. A recent study conducted on critically ill, intubated patients found low ALT to serve as a significant marker for extubation failure, making ALT an easy, yet important laboratory value for identifying such patients [11]. 

A strong association was established between low ALT levels and other accepted parameters determining sarcopenia and frailty status, like the L3SMI score (Striated Muscle Index at the level of L3-vertebra), based on CT imaging analysis of internal medicine patients [12] and the validated FRAIL Questionnaire [13], which also serves for frailty screening and diagnosis. 

Low ALT values are also associated with increased risk of hospitalization, including an increased length of stay and higher rates of re-admission, poor clinical outcomes, and higher mortality among variable populations [9,14] of hospitalized patients following COPD exacerbations, heart failure patients, acute coronary syndrome patients staying in the intensive cardiac care unit (ICCU), and patients suffering from atrial fibrillation. 

### 1.4. Aim of the Current Study

This study aimed to establish, in a retrospective manner, the extent of the potential association of hypothyroidism with sarcopenia and frailty (as indicated by low–normal levels of blood ALT activity) and evaluate the potential negative influences of these comorbidities on patients’ long-term clinical outcomes. 

## 2. Patients and Methods

### 2.1. Study Population

This study population included a sequential 100,000 patients hospitalized in the Chaim Sheba Medical center (the largest tertiary medical center in Israel) from August 2002 to January 2023. Prior to data mining, the study was approved by the Chaim Sheba Institutional Review Board, approval # SMC-23-0087. Informed consent was waived by the IRB due to the retrospective nature of this study. The total number of patients was dictated in an arbitral manner by the software used for data mining. Eligible patients included all those who had both thyroid function tests results and blood ALT activity measurements during their hospital stay. 

To reliably identify sarcopenia and frailty, we restricted the inclusion of patients according to the following rules: We included patients in the age range of 30 to 70 years only, since younger patients are not inclined to suffer from sarcopenia and frailty and older patients will not necessarily benefit from these definitions, as they suffer from frailty accompanying older age. We sought to define the effects of combined sarcopenia, frailty, and hypothyroidism in middle-aged patients. Also, we excluded all patients who had ALT values above the upper limit of normal values (in our institution, values > 40 IU) since such values are arising from lysis of hepatocytes and are commonly indicative of hepatitis and, therefore, such ALT levels are not considered as a reliable parameter of sarcopenia. Also, we excluded patients who had a diagnosis of thyroiditis whose TSH values do not reflect the chronic activity of their thyroid glands, those who had undergone thyroidectomy, and those treated with Amiodarone. The exclusion and partition of the preliminary patients’ cohort to study groups is presented as a CONSORT flow diagram (Figure 1). 

We defined sarcopenia based on the presence of ALT values lower than 12 IU based on several populations’ studies we made in the past—pointing out that the range of sarcopenia definition between 10 IU and 17 IU and the exact value of 12 IU was decided according to its differentiating potential between patients’ sub-populations in this specific study cohort. It should be stated that, in all previous studies, we did not differentiate ALT levels according to gender, since the male-to-female discrepancy in adults is rather small [9,12,13]. 

### 2.2. Biomarker Measurements

Both of the main biomarkers used in this study were not measured for the purpose of clinical research but were used by the Chaim Sheba medical center laboratory for clinical purposes and were analyzed in a retrospective manner. The range of normal ALT and TSH was unchanged during the whole study period and there were no significant methodological changes in the measurement technology over the study period. 

### 2.3. Statistical Analysis

To make sure that our study sample size was sufficient, we calculated for a significance level of 5% to provide the trial with 80% power. We estimated that 5% of the population have TSH values greater than 6. Therefore, to ascertain the sample size, we assumed that the ratio between the groups would be 1:20, while 1 and 20 represent those with abnormal and normal TSH values, respectively. Under the assumption that 15% of patients in the abnormal group and 10% in the normal group would have low ALT levels, we assumed that 15% would have low ALT in the abnormal group and 10% would have low ALT in the normal TSH group. Hence, we calculated a sample size (performed using WINPEPI software, http://www.winepi.net/uk/sample/indice.htm; accessed on 30 March 2024) of 7476 patients to be divided to a minimum of 7120 patients in the normal TSH group, while a minimum of 356 patients would be needed in the low-TSH group of patients.

Categorical variables were described as frequencies and percentages. The distribution of continuous variables was tested by a histogram and a Q-Q plot. Continuous variables that distributed normally are presented as mean ± standard deviations, and those that do not are described as median ± IQR [interquartile range]. A comparison between groups of categorical variables was conducted using a χ^2^ test, and comparison of continuous variables was made using *t*-test for independent samples, or by using a Mann–Whitney test. Multivariable logistic regression was performed to estimate the association between the ALT and the TSH category with adjustment for possible confounders. All the statistical analyses will be two-sided, and *p* < 0.05 will be considered as statistically significant. Statistical analyses were performed using SPSS software version 23. 

## 3. Results

Our retrospective analysis included an overall cohort of 100,000 patients, of whom 16,827 patients were included in the final analysis. We excluded from analysis the following, partially overlapping groups of patients: those over the age of 70 years or below 30 years (for being less relevant when sarcopenia and frailty are looked for: above the age of 70 years, frailty becomes significantly more common, and below the age of 30 years it is rare to non-existent), patients with unavailable values of TSH and/or ALT, patients treated with Amiodarone, and patients diagnosed with thyroiditis or status post-thyroidectomy. It should be stated also that iodine deficiency, as an etiology for hypothyroidism, is almost non-existent in our patient population. Also, we excluded patients with hyperthyroidism (TSH < 0.5 MIU/L) or extreme hypothyroidism (TSH > 20 MIU/L). The total of 16,827 remaining patients included in the analysis were divided into four groups used for comparison of clinical outcomes: patients with normal ALT (robust) and low TSH (euthyroid), those with low ALT (frail) and low TSH (euthyroid), those patients with normal ALT (robust) and high TSH (hypothyroid), and those who were both frail (low ALT) and hypothyroid (high TSH). Table 1 describe these four groups, detailing their demographic and clinical characteristics: as anticipated, the patients that were both frail and hypothyroid were also predominantly females, had lower albumin blood concentrations, and had higher creatinine levels with increased incidence of chronic kidney disease and background malignancies. All the differences reached statistical significance in comparison of the four groups. The following patient characteristics that were not significantly different between our study groups were the frequency of chronic obstructive pulmonary disease (COPD), diabetes mellitus (DM), cerebrovascular accidents, and dementia. 

### 3.1. Univariate Analysis

The patients included in the four groups also differed in terms of their prospects of survival. As detailed in the Kaplan–Meier crude analysis of mortality (Figure 2), the group of frail and hypothyroid (ALT < 12 IU/L and TSH > 6 MIU/L) patients had significantly shorter survival times, while the group of robust and euthyroid (ALT > 12 IU/L and TSH < 6 MIU/L) patients had the longest survival chances. Table 2 includes a univariate analysis relating to the risk of mortality: relative to a median survival duration of 54.19 ± 0.14 months of patients in the reference group (robust and euthyroid, (ALT < 12 IU/L and TSH < 6 MIU/L), the frail and hypothyroid patients survived only for 39.32 ± 2.26 months (*p* < 0.001). 

### 3.2. Multivariate Analysis

A multivariate analysis model (Table 3) included all patient characteristics that had a statistically significant impact on patients’ survival in the univariate analysis. In this model, still, the frail and hypothyroid group of patients (ALT < 12 IU/L and TSH > 6 MIU/L) had a statistically significant increased risk of mortality in the next 5 years (HR = 3.6; CI 2.75−4.71; *p* < 0.001). Patients who were hypothyroid but were considered robust (ALT > 12 IU/L and TSH > 6 MIU/L) were also independently at a higher risk of mortality (1.71 [95% CI 1.44–2.04]) and those patients who were frail and euthyroid (ALT < 12 IU/L and TSH < 6 MIU/L) had a higher risk of mortality (HR = 1.39, 95% CI 1.24–1.56). Other patient characteristics that were found to be independently associated with increased risk of mortality in a multivariate analysis were chronic obstructive pulmonary disease (increased by 34%), chronic kidney disease (increased by 153%), diabetes mellitus (increased by 37%), active malignancy (increased by 312%), and concurrent dementia (increasing the risk of mortality by 89%). 

## 4. Discussion

It seems that even though the thyroid gland’s importance and functions have been established for decades, the full spectrum of impact of thyroid hormones on the human tissues’ cells has yet to be fully elucidated. In their review, Mandoza and Hollenberg detail novel mechanisms by which the thyroid hormones’ activities differentiate according to receptor functions and intracellular modulations of different body tissue cells [15]. In his editorial relating to thyroid functions and longevity, Peeters describes the increasing TSH levels as people get older and the difficulties in genuine assessment of thyroid functionality in the face of accumulating chronic diseases and elongating lists of medications. Nevertheless, although he concludes that decreased thyroid function is part of getting old, he does not associate this phenomenon with either sarcopenia or frailty [16]. 

Indeed, hypothyroidism is a common phenomenon in both community-residing patients and hospitalized patients as well, as they get older. Past research has associated hypothyroidism with a myriad of pathologies and shortened survival [17,18]. In reviews that do not concentrate on mortality, the disability associated with hypothyroidism is emphasized [19]. In their narrative review, Ettelson and Papaleontio extend the therapeutic targets for hypothyroid patients: beyond normalization of TSH values, they encourage clinicians to peruse better quality-of-life outcomes using the PROMs, patient-reported outcome measures [20]. Their attitude towards relating to hypothyroidism as a multisystem disease, rather than a disease of “a gland,” aligns with the strong association between hypothyroidism, sarcopenia, and frailty. In a publication by Lee et al. [21] relying on data retrieved from the Korea National Health and Nutrition Examination Survey (2013 to 2015), a statistically significant correlation was found between hypothyroidism, both clinical and sub-clinical, and frailty. In patients with frank hypothyroidism, they found a correlation between TSH levels and the relative risk of development of frailty. Their frailty diagnosis relied on the Fried frailty phenotype and, therefore, could not apply to large populations by simply screening their health-related characteristics. This is not the case when screening for sarcopenia and frailty, as these conditions are identified based on low ALT blood levels, data that are readily available in patients who have undergone routine blood tests in the past six months. 

Low ALT values, when used as a biomarker for sarcopenia and frailty, have already been shown to be associated with shortened survival of diverse patient populations, as presented above. The association of diagnosing both hypothyroidism and frailty via simple blood testing is indeed tempting for clinicians and enables sarcopenia diagnosis that is beyond the old “eyeballing” methods of assessment. 

In the current study, we performed a retrospective analysis of hospitalized patients regarding their hypothyroidism status (according to past TSH measurements) and, simultaneously, their status of being either robust or sarcopenic and frail (according to their concurrent ALT measurements). Sarcopenia and frailty are also known to be associated with increased risk of falls, hospitalizations [9], and mortality [9,10,14,22]. The combination of both disease states, aided by low ALT values as a biomarker for sarcopenia and frailty, was never questioned, as far as we know, as to the extent it would affect patients’ long-term prognosis. 

Alongside past measurements of TSH levels as an indicator of hypothyroidism, we relied on several recent publications, showing that low ALT levels could serve as a retrospective biomarker for sarcopenia and frailty [10,14,23]. We grouped our patients into four groups according to their TSH and ALT levels (robust and euthyroid, robust and hypothyroid, frail and euthyroid, and frail and hypothyroid) and found that the combination of hypothyroidism and frailty is, indeed, associated with significantly shortened survival. 

Modern views of personalized medicine are concentrating on patients as a whole. This “gestalt” view of patients is complementary to the precise medicine strategies which concentrate on the disease state rather than the patients. Our findings suggest that the combination of hypothyroidism and frailty should help clinicians to better tailor both diagnostic and therapeutic approaches to their patients. For example, whenever there is a doubt regarding the need to address sub-clinical hypothyroidism with supplementary levothyroxine, clinicians should obtain a complete “picture” of their patient by measuring their ALT levels. If these levels are suggestive of sarcopenia and frailty (ALT < 12 IU/L), it would only be prudent to initiate supplemental levothyroxine. The opposite is also true: there are few means, currently, to reverse sarcopenia and frailty. Nevertheless, these patients should be screened in a timely manner for a potential, so-called, sub-clinical hypothyroidism that should be addressed pharmacologically. 

## 5. Conclusions

Clinicians should be aware of the relatively common co-existence of hypothyroidism and sarcopenia. These comorbidities should be considered as significantly damaging patients’ quality of life and, at times, shortening patients’ survival, and therefore be promptly addressed—either by correcting thyroid function tests/reversing etiologies of acquired hypothyroidism or addressing potential causes for sarcopenia and frailty such as poorly controlled diabetes mellitus. The main contribution of our work is the fact that both morbidities were diagnosed by readily available laboratory tests. Both low ALT and increased TSH levels could be monitored in middle-aged men and women, such as our cohort of patients. 

### Limitations

This was a retrospective, single-center study and, as such, should be followed by long-term, prospective interventional studies. The results should be generalized with caution regarding other patient populations. Causes of hypothyroidism are not detailed in this manuscript. We excluded some, but it is possible that certain causes of hypothyroidism might be directly associated with sarcopenia and frailty. This possibility cannot be ruled out based on our study results. 

## Figures and Tables

**Figure 1 jcm-13-05838-f001:**
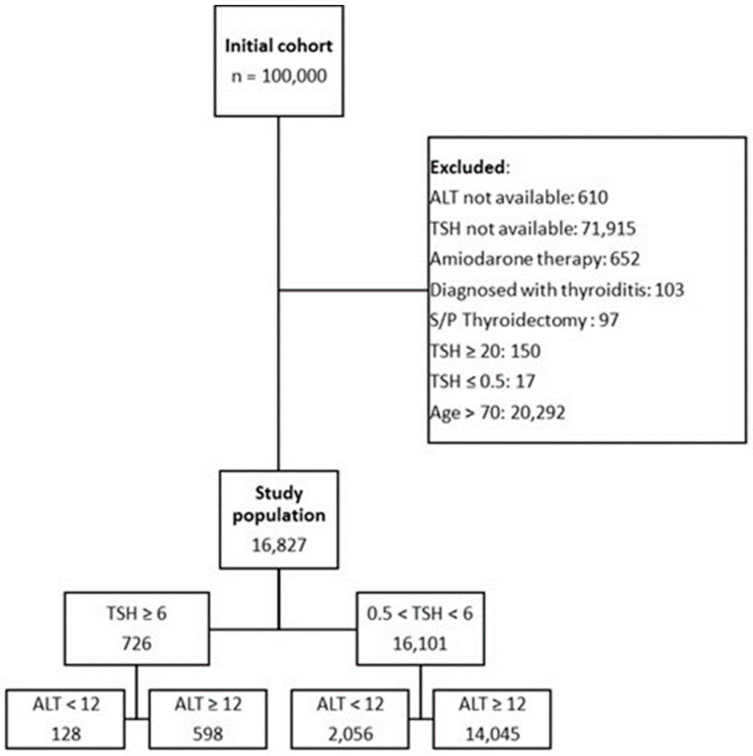
CONSORT flow diagram. Described is the flow of patients from initial recruitment (ALT levels not exceeding 40 IU) through omission of patients due to exclusion criteria and the final division of patients according to their ALT and TSH values.

**Figure 2 jcm-13-05838-f002:**
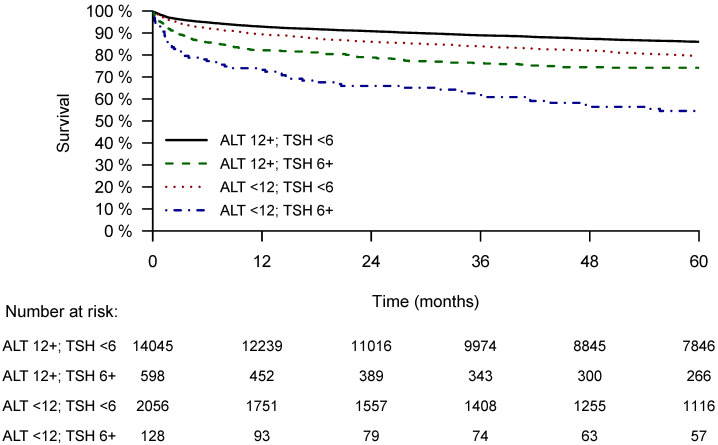
Survival analysis according to frailty and hypothyroidism status. The Kaplan–Meier survival function separating four patients’ groups according to their ALT and TSH values: the ongoing, significant differences in their survival probabilities are shown along the time axis.

**Table 1 jcm-13-05838-t001:** Patients’ baseline characteristics according to TSH and ALT blood activity level.

	Total Cohort *n* = 16,827	ALT ≥ 12, TSH < 6 *n* = 14,045	ALT < 12, TSH < 6 *n* = 2056	ALT ≥ 12, TSH ≥ 6 *n* = 598	ALT < 12, TSH ≥ 6 *n* = 128	*p* Value
Demographics						
Age (years) (IQR)	56.9 (46.5–64)	57 (47–63.9)	55 (41.8–63.8)	59 (50.3–64.8)	56.2 (45.5–64.3)	<0.001
Male, *n* (%)	9507 (56.5)	8394 (59.8)	821 (39)	249 (41.6)	43 (33.6)	<0.001
Comorbidities						
Hypertension, *n* (%)	5479 (32.6)	4687 (33.4)	586 (28.5)	172 (28.8)	34 (26.6)	<0.001
IHD, *n* (%)	3325 (19.8)	2933 (20.9)	272 (13.2)	108 (18.1)	12 (9.4)	<0.001
Dyslipidemia, *n* (%)	2417 (14.4)	2125 (15.1)	208 (10.1)	73 (12.2)	11 (8.6)	<0.001
COPD, *n* (%)	831 (4.9)	675 (4.8)	114 (5.5)	30 (5)	12 (9.4)	0.057
CKD, *n* (%)	1051 (6.4)	775 (5.6)	194 (9.7)	55 (9.4)	27 (21.4)	<0.001
Diabetes, *n* (%)	3589 (21.3)	3004 (21.4)	433 (21.1)	124 (20.7)	28 (21.9)	0.96
Malignancy, *n* (%)	2966 (17.6)	2386 (17)	405 (19.7)	139 (23.2)	36 (28.1)	<0.001
Stroke, *n* (%)	2639 (15.7)	2233 (15.9)	313 (15.2)	80 (13.4)	13 (10.2)	0.1
Dementia, *n* (%)	255 (1.5)	202 (1.4)	44 (2.1)	6 (1)	3 (2.3)	0.056
Laboratory parameters						
Albumin (g/dL); (Mean ± SD)	3.8 ± 0.5	3.8 ± 0.5	3.6 ± 0.5	3.6 ± 0.6	3.4 ± 0.6	<0.001
Creatinine (mg/dL); (Mean ± SD)	0.9 ± 0.6	0.94 ± 0.5	1 ± 1	1 ± 0.7	1.3 ± 1.1	<0.001

Continuous variables are presented as the median (interquartile range) or mean ± standard deviation. Categorical variables are presented as numbers (percentage); ALT—alanine aminotransferase; TSH—thyroid-stimulating hormone (MIU/L); IHD—ischemic heart disease; COPD—chronic obstructive pulmonary disease; CKD—chronic kidney disease; chronic kidney disease defined as serum creatinine level above 1.5 mg/dL.

**Table 2 jcm-13-05838-t002:** Survival according to frailty and hypothyroidism status, a univariate analysis.

Patients Grouping	Months	*p* Value
Reference group (ALT > 12, TSH < 6)	54.19 ± 0.14	<0.001
Group 1 (ALT > 12, TSH > 6)	47.35 ± 0.94	
Group 2 (ALT < 12, TSH < 6)	51.48 ± 0.42	
Group 3 (ALT < 12, TSH > 6)	39.32 ± 2.26	
Whole cohort	53.5 ± 0.13	

ALT: alanine aminotransferase; TSH: thyroid-stimulating hormone, MIU/L; data are presented as months ± standard deviation.

**Table 3 jcm-13-05838-t003:** Multivariate analysis for all variables with potential impact on patients’ survival in a univariate analysis.

Patient Characteristics	Hazard Ratio	*p* Value
Group 1 (ALT > 12 IU/L, TSH > 6 MIU/L)	1.71 [1.44–2.04]	<0.001
Group 2 (ALT > 12 IU/L, TSH > 6 MIU/L)	1.39 [1.24–1.56]	<0.001
Group 3 (ALT > 12 IU/L, TSH < 6 MIU/L)	3.6 [2.75–4.71]	<0.001
Age (years)	1.04 [1.03–1.05]	<0.001
Gender (male)	0.85 [0.78–0.93]	<0.001
Hypertension	0.7 [0.64–0.77]	<0.001
Ischemic heart disease	0.82 [0.73–0.92]	<0.001
Dyslipidemia	0.64 [0.56–0.73]	<0.001
Chronic obstructive pulmonary disease	1.34 [1.16–1.54]	<0.001
Chronic kidney disease	2.53 [2.25–2.83]	<0.001
Diabetes mellitus	1.37 [1.25–1.51]	<0.001
Malignancy	4.12 [3.79–4.48]	<0.001
Stroke	0.79 [0.7–0.89]	<0.001
Dementia	1.89 [1.53–2.35]	<0.001

## Data Availability

Data will be available from the principal investigator in accordance with the IRB regulations.

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
