# Peer review of "Comorbid Hypothyroidism and Low-Alanine Aminotransferase-Associated Sarcopenia Associated with Shortened Survival: A Retrospective Study of 16,827 Patients over a 21-Year Period"

_jcm, 2024, doi:10.3390/jcm13195838_

Round 1

Reviewer 1 Report

Comments and Suggestions for Authors

The authors retrospectively investigated the relationship of hypothyroidism with sarcopenia and frailty in hospitalized patients.

Overall the paper is well written. Here are my recommendations for improvement.

1) The abstract should have an explicitly stated objective statement.

2) Introduction: They should state the aim very precisely. There is no need to mention what and why they did not do. This can be discussed in the discussion if they desire. Just say what you set out to accomplish.

3) Although they have stated the ethics in the acknowledgment section, it should also be mentioned in the methods, under the subjects sub-section.

4) Authors should justify the use of ALT 12 IU/L as a cut-off value. The ALT clinical cut-off values are different for men and women. Yet they combined men and women in the analysis.

5) Authors should do the multivariate analysis after adjusting the analysis for gender and age because these are important determinants of frailty or ALT concentrations.

6) Authors have stated that they have performed multivariate analysis but they failed to mention what are the "multi variables' used in this study.

7) The authors should explain the methods used for biomarkers studies in this study.

8) They have included subjects from 2022 through 2023. They have not reported any methodological changes that may have occurred. This is a very important factor to consider in using retrospective data from the past. Generally, methods get better and better over time. The ALT values in 2023 may not be compatible with those in 2002. This needed to be discussed and stated as a potential limitation and its impact on the findings.

9) Tables are stand-alone and self-sustaining. That means a reader should understand the data presented in the table without referring to the text of the manuscript. So, please add more footnotes at the bottom of the table with appropriate superscripts embedded in the text of the table. Also, the table footnote should contain a type of statistical test used, abbreviations used, whether the data were mean ± SD or SE, and the significance level.

10) Like tables, figures are also stand-alone and self-sustaining. That means a reader should understand the data presented in the figure without referring to the text of the manuscript. So, please add a description after the title of the figure. This description should contain (not be limited to) the interpretation of data, the type of statistical test used, abbreviations used, whether the data were mean ± SD or SE, and the significance level.

11) "Levels" should be changed to "concentrations" throughout the paper. Level is not the right term to use.

12) Authors have stated that they have excluded subjects with higher ALT values (greater than 40 IU/L), but in Figure 1, it is not stated.

Comments on the Quality of English Language

Minor editing is needed.

Reviewer 2 Report

Comments and Suggestions for Authors

A retrospective investigation on the comorbidity of low-alanine aminotransferase-associated sarcopenia and hypothyroidism was given by the authors. A complicated interaction of hormonal and metabolic variables leads to the comorbidity of hypothyroidism and low-alanine aminotransferase (ALT)-associated sarcopenia. Studies show a strong correlation between sarcopenia and low ALT levels, especially in the elderly and those with chronic illnesses. Low ALT is linked to frailty and a lower survival time in cancer patients, suggesting that it may be a marker for sarcopenia. The authors came to the conclusion that frailty and sarcopenia are frequent comorbidities with poor long-term clinical outcomes. Low ALT levels may be a helpful indicator for individuals who have already received a hypothyroidism diagnosis confirmg findings in oncological patients (10.3390/cancers16010174).

Please specify the innovative content of the present paper, thanks.

Round 2

Reviewer 1 Report

Comments and Suggestions for Authors

They have addressed my recommendations